# PROSE: Predicting Operators and Symbolic Expressions using Multimodal Transformers

## Abstract

Approximating nonlinear differential equations using a neural network provides a robust and efficient tool for various scientific computing tasks, including real-time predictions, inverse problems, optimal controls, and surrogate modeling. Previous works have focused on embedding dynamical systems into networks through two approaches: learning a single solution operator (i.e., the mapping from input parametrized functions to solutions) or learning the governing system of equations (i.e., the constitutive model relative to the state variables). Both of these approaches yield different representations for the same underlying data or function. Additionally, observing that families of differential equations often share key characteristics, we seek one network representation across a wide range of equations. Our method, called **Pr**edicting **O**perators and **S**ymbolic **E**xpressions (PROSE), learns maps from multimodal inputs to multimodal outputs, capable of generating both numerical predictions and mathematical equations. By using a transformer structure and a feature fusion approach, our network can simultaneously embed sets of solution operators for various parametric differential equations using a single trained network. Detailed experiments demonstrate that the network benefits from its multimodal nature, resulting in improved prediction accuracy and better generalization. The network is shown to be able to handle noise in the data and errors in the symbolic representation, including noisy numerical values, model misspecification, and erroneous addition or deletion of terms. PROSE provides a new neural network framework for differential equations which allows for more flexibility and generality in learning operators and governing equations from data.

## 1 Introduction

Differential equations are important tools for understanding and studying nonlinear physical phenomena and time-series dynamics. They are necessary for a multitude of modern scientific and engineering applications, including stability analysis, state variable prediction, structural optimization, and design. Consider parametric ordinary differential equations (ODEs), i.e. differential equations whose initial conditions and coefficients are parameterized by functions with inputs from some distribution. We can denote the system by $\frac{d\boldsymbol{u}}{dt} = f(\boldsymbol{u}; a_s(t))$, where $\boldsymbol{u}(t) \in \mathbb{R}^d$ are states, and $a_s(t)$ is the parametric function with input parameter $s$. For example, $a_s(t)$ could be an additive forcing term where $s$ follows a normal distribution. The goal of computational methods for parametric ODEs is to evaluate the solution given a new parametric function, often with the need to generalize to larger parameter distributions, i.e. out-of-distribution predictions.

Recently, *operator learning* has been used to encode the operator that maps input functions $a_s(-)$ to the solution $\boldsymbol{u}(-; a_s(-))$ through a deep network, whose evaluation is more cost-efficient than fully simulating the differential equations (Chen & Chen, 1995; Li et al., 2020; Lu et al., 2021; Lin et al., 2021; Zhang et al., 2023a). An advantage of operator learning compared to conventional networks is that the resulting approximation captures the mapping between functions, rather than being limited to fixed-size vectors. This flexibility enables a broader range of downstream tasks to be undertaken, especially in multi-query settings. However, operator learning is limited to training solutions for an individual differential equation. In particular, current operator learning methods do not benefit from observations of similar systems and, once trained, do not generalize to new differential equations.

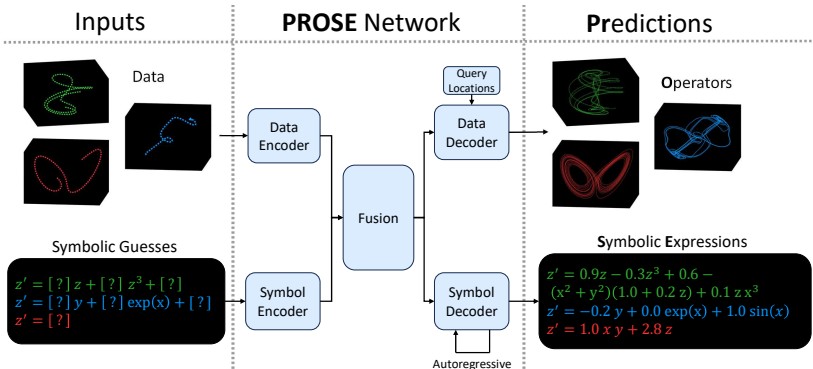

Figure 1: **PROSE network illustration.** The inputs and outputs (predictions) are multimodal, each including numerical values (data) and symbolic expressions (governing equations). Here we include just the third term in the governing equations for simpler visualization.

**Problem Statement**   We consider the problem of encoding multiple ODEs and parametric functions, for use in generalized prediction and model discovery. Specifically, we are given $N$ ODEs $f_j$, and parametric functions $a_s^j(t)$, with the goal of constructing a single neural network to both identify the system and the operator from parametric functions $a_s^j(-)$ to solutions. Consider a family of differential equations indexed by $j = 1, \cdots, N$, with the form $\frac{d\boldsymbol{u}}{dt} = f_j\left(\boldsymbol{u}; a_s^j(t)\right)$, where the solutions are denoted by $\boldsymbol{u}_j(-; a_s^j(-))$. The solution operator $G^j$ encodes the solution's dependence on $a_s^j$ and corresponds to the $j^{\text{th}}$ ODE. When utilizing standard operator learning, it becomes necessary to train separate deep networks for each of the $N$ equations. That approach can quickly become impractical and inefficient, especially in the context of most nonlinear scientific problems.

This work introduces a multimodal framework for simultaneously encoding multiple operators for use in predicting states at query locations and discovering the governing model that represents the equations of motion describing the data. For data prediction, a novel transformer-based approach which we call *multi-operator learning* is employed. This entails training the network to learn the solution operator across a set of distinct parametric dynamical systems. In other words, the network learns a single operator $\bar{G}$ that represents the family of mappings $\left\{G^1, \cdots, G^N\right\}$ by leveraging shared characteristics among their features. This should also allow the network to predict new operators that share commonalities with those from the family of operators used in training, i.e. generalize to new operators. During testing or prediction, the governing equations (i.e. the mathematical equations defining the dynamics of dependent variables for a given data sequence) are not known, so the algorithm also produces a symbolic expression using a generative model. In other words, the network learns a syntax for representing and articulating differential equations. In this way, the approach yields a network capable of evaluating dependent variables at query locations over wide parameter sets and also "writes" the mathematical differential equation associated to the data. This can be viewed as a large language model for differential equations.

**Main Contributions**   The Predicting Operators and Symbolic Expression (PROSE) framework introduces a new approach to learning differential equations from data. The key components of the architecture are illustrated are Figure 1. The main contributions and novelty are summarized below.

- PROSE is the first method to generate both the governing system and an operator network from multiple distinct ODEs. It is one of the first multi-operator learning approaches.

- PROSE incorporates a new modality through a fusion structure. Unlike text modality or labels, the symbolic expression can accurately generate the system solution.

- The network architecture introduces new structural elements, including a fusion transformer that connects the data and embedded symbols.

- We demonstrate accuracy in generating valid ODEs (validity is of $> 99.9\%$ on in-distribution tests and $> 97.89\%$ on out-of-distribution predictions), showing that PROSE can generate new ODEs from data.

## 2 RELATED WORKS

PROSE is both a multi-operator learning and a model discovery approach. We summarize these two distinct research areas in this section.

**Operator Learning** Operator learning (Chen & Chen, 1993; 1995; Li et al., 2020; Lu et al., 2021; Lin et al., 2021; Zhang et al., 2023a) studies neural network approximations to an operator $G : U \to V$, where $U$ and $V$ are function spaces. This approach holds significant relevance in various mathematical problems, including the solution of parametric PDEs (Bhattacharya et al., 2020; Kovachki et al., 2021), control of dynamical systems (Lin et al., 2022; Yeh et al., 2023), and multi-fidelity modeling (Lu et al., 2022b; Zhang et al., 2023b; Ahmed & Stinis, 2023). Operator learning has gained substantial popularity within the mathematical and scientific machine learning community, with applications in engineering domains (Pathak et al., 2022). Currently, methods for neural operators focus on constructing a single operator, e.g. learning the map from the initial conditions or parameters of a physical system to the solution at a terminal time.

In Chen & Chen (1993; 1995), the authors extended the universal approximation theory from function approximation (Cybenko, 1989; Jones, 1992; Barron, 1993) to operators. This work paved the way for the modern development of deep neural operator learning (DON) as seen in Lin et al. (2021); Lu et al. (2021; 2022a). Building upon the principles of Chen & Chen (1995), Zhang et al. (2023a) further expanded this approach by constructing operator networks that remain invariant to the input/output function discretizations. The noisy operator learning and optimization is studied in Lin et al. (2021). Another operator approach is the Fourier neural operators (FNO) (Li et al., 2020; Wen et al., 2022), which use Fourier transformations and their inverses in approximating operators through kernel integral approximations. Comparative analysis can be found in Lu et al. (2022a); Zhang et al. (2023a).

The multi-input-output network (MioNet) (Jin et al., 2022) extends operator learning to handle multiple input/output parametric functions within the single operator framework. Recently, the In-Context Operator Network (ICON) (Yang et al., 2023a) was developed for multi-operator learning using data and equation labels (one-hot encoding) as prompts and a test label during inference. This was later extended to include multimodal inputs by allowing captions which are embedded into the input sequence using a pre-trained language model (Yang et al., 2023b). Multi-operator learning has significant challenges, especially when encoding the operators or when addressing out-of-distribution problems (i.e. those that extend beyond the training dataset).

**Learning Governing Equations** Learning mathematical models from observations of dynamical systems is an essential scientific task, resulting in the ability to analyze relations between variables and obtain a deeper understanding of the laws of nature. In the works Bongard & Lipson (2007); Schmidt & Lipson (2009), the authors introduced a symbolic regression approach for learning constitutive equations and other physically relevant equations from time-series data. The SINDy algorithm, introduced in Brunton et al. (2016), utilizes a dictionary of candidate features that often includes polynomials and trigonometric functions. They developed an iterative thresholding method to obtain a sparse model, with the goal of achieving a parsimonious representation of the relationships among potential model terms. SINDy has found applications in a wide range of problems and formulations, as demonstrated in Kaiser et al. (2018); Champion et al. (2019); Rudy et al. (2019); Hoffmann et al. (2019); Shea et al. (2021); Messenger & Bortz (2021). Sparse optimization techniques for learning partial differential equations were developed in Schaeffer (2017) for spatio-temporal data. This approach incorporates differential operators into the dictionary, and the governing equation is trained using the LASSO method. The $\ell^1$-based approaches offer statistical guarantees with respect to the error bounds and equation recovery rates. These methods have been further refined and extended in subsequent works, including Schaeffer & McCalla (2017); Schaeffer et al. (2017; 2018; 2020); Liu et al. (2023). In Chen et al. (2021), the Physics-Informed Neural Network with Sparse Regression (PINN-SR) method for discovering PDE models demonstrated that the equation learning paradigm can be leveraged within the PINNs (Raissi et al., 2019; Karniadakis et al., 2021; Leung et al., 2022) framework to train models from scarce data. The operator inference technique (Peherstorfer & Willcox, 2016) approximates high-dimensional differential equations by first reducing the data-dimension to a small set of variables and training a lower-dimensional ODE model using a least-squares fit over polynomial features. This is particularly advantageous when dealing

with high-dimensional data and when the original differential equations are inaccessible. More recently, transformer-based sequence-to-sequence models have been proposed to perform symbolic regression and scalar autonomous ODE model discovery from numerical inputs only (Lample & Charton, 2020; Becker et al., 2023) (i.e. data to symbol maps).

## 3 METHODOLOGY

The main ingredients of PROSE include symbol embedding, transformers, and multimodal inputs and outputs. We summarize these key elements in this section.

**Transformers**  A transformer is an attention-driven mechanism that excels at capturing longer-term dependencies in data (Vaswani et al., 2017; Dai et al., 2019; Beltagy et al., 2020). The vanilla transformer uses a self-attention architecture (Bahdanau et al., 2014; Xu et al., 2015), enabling it to capture intricate relationships within lengthy time series data. Specifically, let us denote the input time series data as $X \in \mathbb{R}^{n \times d}$, where $n$ is the number of time steps and $d$ is the dimension of each element in the time series. Self-attention first computes the projections: query $Q = XW^Q$, key $K = XW^K$ and value $V = XW^V$, where $W^Q \in \mathbb{R}^{d \times d_k}$, $W^K \in \mathbb{R}^{d \times d_k}$, and $W^V \in \mathbb{R}^{d \times d_v}$. It then outputs the context $C \in \mathbb{R}^{n \times d_v}$ via $C = \text{softmax}\left(\frac{QK^T}{\sqrt{d_k}}\right) V$, where the softmax function is calculated over all entries of each row. Self-attention discovers relationships among various elements within a time sequence. Predictions often depend on multiple data sources, making it crucial to understand the interactions and encode various time series data (see Section 3 for details). This self-attention idea has driven the development of the cross-attention mechanism (Lu et al., 2019; Tsai et al., 2019; Li et al., 2021). Given two input time series data $X, Y$, cross-attention computes the query, key, and value as $Q = XW^Q$, $K = YW^K$, and $V = YW^V$. In the case where $Y$ represents the output of a decoder and $X$ represents the output of an encoder, the cross-attention, which directs its focus from $X$ to $Y$, is commonly referred to as encoder-decoder attention (Vaswani et al., 2017). Encoder-decoder attention serves as a crucial component within autoregressive models (Graves, 2013; Vaswani et al., 2017; Li et al., 2021). The autoregressive model operates by making predictions for a time series iteratively, one step at a time. To achieve this, it utilizes the previous step's generated output as additional input for the subsequent prediction. This approach has demonstrated the capacity for mitigating accumulation errors (Floridi & Chiriatti, 2020), which makes it desirable for longer-time predictions.

**Multimodal Machine Learning**  Multimodal machine learning (MML) trains models using data from heterogeneous sources (Lu et al., 2019; Sun et al., 2019; Tan & Bansal, 2019; Li et al., 2021; Xu et al., 2023). Of major interest in this topic are methods for the fusion of data from multiple modalities, the exploration of their interplay, and the development of corresponding models and algorithms. For instance, consider the field of visual-language reasoning (Tan & Bansal, 2019; Sun et al., 2019; Li et al., 2019), where the utilization of visual content, such as images or videos, with the semantics of language (Tan & Bansal, 2019) associated with these visual elements, such as captions or descriptions, leads to the development of models with richer information (Li et al., 2019). Another illustrative example is that of AI robots, which use multimodal sensors, including cameras, radar systems, and ultrasounds, to perceive their environment and make decisions (Feng et al., 2020; Liu et al., 2021). In mathematical applications, researchers employ multiscale mathematical models (Efendiev et al., 2022), where each modality is essentially characterized by varying levels of accuracy, to train a single model capable of predicting multiscale differential equations effectively.

**Operator Learning Structure**  The authors in Chen & Chen (1995) established a universal approximation theory for continuous operators, denoted by $G$. Particularly, they showed that the neural operator $G_\theta(u)(t) = \sum_{k=1}^{K} b_k(t) p_k(\hat{u})$ can approximate $G(u)(t)$ for $t$ in the output function domain (under certain conditions). Here $p(\cdot)$ and $b(\cdot)$ are neural networks which are called the branch and trunk (Lu et al., 2021), and $\hat{u}$ is a discretized approximation to the input function $u$. In our applications, these input functions $u$ correspond to ODE solutions sampled in the input intervals, and the output functions are solutions over larger intervals. Based on the output-discretization invariance property of the network (Lu et al., 2022a; Zhang et al., 2023a), the output of the operator network can be queried at arbitrary timepoints, allowing predictions of the solution at any location.

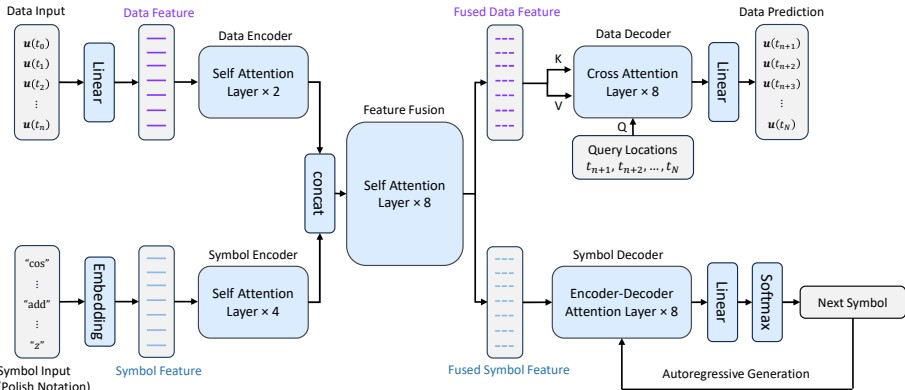

Figure 2: **PROSE architecture and the workflow.** Data Input and Symbol Input are embedded into Data Feature and Symbol Feature respectively before encoding and fusion through Feature Fusion. PROSE uses Cross-Attention to construct the operator (upper-right structure) from Fused Data Feature, and evaluate it at Query Locations. PROSE generates symbolic expressions in the lower-right portion autoregressively. Attention blocks are displayed in Appendix D, where each layer also includes a feedforward network.

**Equation Encoding via Polish Notation**  Mathematical expressions can be encoded as trees with operations and functions as nodes, and constants and variables as leaves (Liang & Yang, 2022; Jiang et al., 2023). For instance, the tree on the right represents the expression $\cos(1.5x_1) + x_2^2 - 2.6$.

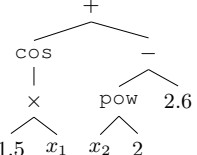

Trees provide natural evaluation orders, eliminating the need to use parentheses or spaces. Under some additional restrictions (e.g. $1 + 2 + 3$ should be processed as $1 + (2 + 3)$, $-1 \times x$ is equivalent to $-x$), there is a one-to-one correspondence between trees and mathematical expressions. For these reasons, trees provide an unambiguous way of encoding equations. While there are existing tree2tree methods (Tai et al., 2015; Dyer et al., 2016), they are usually slower than seq2seq methods at training and inference time. The preorder traversal is a consistent way of mapping trees to sequences, and the resulting sequences are known as Polish or Prefix notation (Pogorzelski, 1965), which is used in our equation encoder. For the above expression $\cos(1.5x_1) + x_2^2 - 2.6$, its Polish notation is given by the sequence `[+ cos × 1.5 `$x_1$` − pow `$x_2$` 2 2.6]`. Operations such as `cos` are treated as single words and are not further tokenized, but they are trainable. In comparison to LaTeX representations of mathematical expressions, Polish notations have shorter lengths, simpler syntax, and are often more consistent. Note that in Liang & Yang (2022); Jiang et al. (2023), binary trees of depth-3 are used to generate symbolic approximations directly for the solution of a single differential equation.

Following Charton (2022); d'Ascoli et al. (2022); Kamienny et al. (2022), to have a reasonable vocabulary size, floating point numbers are represented in base-10 notations, each consisting of three components: sign, mantissa, and exponent, which are treated as words with trainable embedding. For example, if mantissa length is chosen to be 3, then $2.6 = +1 \cdot 260 \cdot 10^{-2}$ is represented as `[+ 260 E-2]`. For vector-valued functions, a dimension-separation token is used, i.e. $\boldsymbol{f} = (f_1, f_2)$ is represented as "$f_1 \mid f_2$". Similar to Charton (2022); d'Ascoli et al. (2022); Kamienny et al. (2022), we choose mantissa length 3, resulting in a vocabulary of size about 1,100.

## 3.1 MODEL OVERVIEW

Our network uses hierarchical attention for feature processing and fusion, and two transformer decoders for two downstream tasks. Figure 2 provides an overview of the architecture. The PROSE architecture contains five main components trained end-to-end: data encoder, symbol encoder, feature fusion, data decoder, and symbol decoder.

**Encoders**  Two separate transformer encoders are used to obtain domain-specific features. Given numerical data inputs and symbolic equation guesses (possibly empty or erroneous), the data en-

coder and symbol encoder first separately perform feature aggregation using self-attention. For a data input sequence $\boldsymbol{u}(t_0), \cdots, \boldsymbol{u}(t_n)$, each element $\boldsymbol{u}(t_i)$, together with its time variable $t_i$, goes through a linear layer to form the Data Feature (purple feature sequence in Figure 2). PROSE then uses self-attention to further process the Data Feature, where the time variables $t_i$ serve as the positional encoding. The symbolic input (in Polish notation) is a standard word sequence, which can be directly processed with self-attention layers. The word embedding (for operations, sign, mantissa, etc.) is randomly initialized and trainable. Sinusoidal positional encoding (Vaswani et al., 2017) is used for the symbol encoder.

**Feature Fusion**   Hierarchical attention (multi-stream to one-stream) is used in this model for feature fusion. Separately-processed data and symbol features are concatenated into a feature sequence, and further processed through self-attention layers where modality interaction occurs. Following Kim et al. (2021), a learnable modality-type embedding is added to the fused features, explicitly signaling to the model which parts of the sequence are from which modality. Positional encoding is not needed since it is already included in the individual encoders.

**Data Decoder**   The data decoder constructs the operator via the cross-attention mechanism, establishing a link between the input-encoded time sequence (fused data features) and the output functions. The query locations, representing the independent variables of these output functions, serve as the evaluation points. Importantly, these query locations operate independently of each other, meaning that assessing the operator at one point, $t_i$, does not impact the evaluation of the operator at another point, $t_j$. As a result, the time and space complexity scales linearly with the number of query locations. In addition, since the evaluation points are independent of the network generation, this resembles the philosophy of the branch and trunk nets, see Operator Learning Structure in Section 3.

**Symbol Decoder**   The symbol decoder is a standard encoder-decoder transformer, where the fused symbol feature is the context for generation. The output equation is produced using an autoregressive approach (Vaswani et al., 2017; Floridi & Chiriatti, 2020): it starts with the start-of-sentence token and proceeds iteratively, generating each term of the equation based on prior predictions, until it encounters the end-of-sentence token for that specific equation. During evaluation time, greedy search (iterative selection of symbol with maximum probability) is used for efficient symbol generation. While beam search (Wu et al., 2016) can be used to improve the performance (e.g. percentage of valid expression outputs), we empirically find that greedy search is sufficient for obtaining valid mathematical expressions using the Polish notation formulation.

## 4 EXPERIMENTS

We detail the numerical experiments and studies in this section. We created a dataset of 15 distinct multi-dimensional nonlinear ODEs. To verify the performance of the PROSE approach, we conduct four case studies (Table 2) with different symbolic and data inputs (Table 1). Additionally, in the ablation study, we confirm that the inclusion of symbolic equation information enhances the accuracy of the data prediction. Hyperparameters and experimental conditions can be found in Appendix A.

**Dataset**   The dataset is created from a dictionary of 15 distinct ODEs with varying dimensions: twelve 3D systems, two 4D systems, and one 5D system. To generate samples, we uniformly sample the coefficients of each term in the ODEs from the range $[F - 0.1F, F + 0.1F]$, where $F$ represents the value of interest. We refer to Appendix B for the details.

The goal is to accurately predict the solutions of ODEs at future timepoints only using observations of a few points along one trajectory. We do not assume knowledge of the governing equation and thus the equations are also trained using the PROSE approach. The operator's input function is the values along the trajectories, discretized using a 64-point uniform mesh in the interval $[0, 2]$. The target operator maps this input function to the ODE solution in the interval $[2, 6]$. To assess PROSE's performance under noisy conditions, we introduce $2\%$ Gaussian noise directly to the data samples.

The training dataset contains 512K examples, where 20 initial conditions are sampled to generate solution curves for each randomly generated system. The validation dataset contains 25.6K examples, where 4 initial conditions are sampled for each ODE system. The testing dataset contains 102,400 examples, where 4 initial conditions are sampled for each ODE system. The training dataset and the

Table 1: **Experiment settings.** Data-noise: additive noise on data. Unknown coefficients: replace the input equation coefficients with placeholders. Term deletion: omit a term in the target equation with $15\%$ chance. Term addition: add an erroneous term with $15\%$ chance. For the last test, all data inputs are padded to the maximum equation dimension. "Unknown expressions" means that the coefficients are unknown and there are terms added and removed.

| Experiments (Expression Type) | Data-Noise | Unknown Coefficients | Term Deletion | Term Addition | # ODEs |
|---|---|---|---|---|---|
| Known | ✓ | ✗ | ✗ | ✗ | 12 |
| Skeleton | ✓ | ✓ | ✗ | ✗ | 12 |
| Unknown (3D) | ✓ | ✓ | ✓ | ✓ | 12 |
| Unknown (Multi-D) | ✓ | ✓ | ✓ | ✓ | 15 |

testing dataset contain the same number of ODE systems. In terms of practical applications, given test cases with unknown models, we are free to continue to augment the training and validation sets with any ODE, thus the dataset can be made arbitrarily large.

To test the performance of the equation prediction, we corrupt the input equation by randomly replacing, deleting, and adding terms. The terminologies and settings are found in Table 1.

**Evaluation Metrics**   As PROSE predicts the operator and learns the equation, we present three metrics to evaluate the model performance for solution and equation learning. For data prediction, the relative $L^2$ error is reported. For the expression outputs (symbolic sequences in Polish notation), a decoding algorithm is used to transform the sequences into trees representing functions. The percentage of outputs that can be transformed into valid mathematical expressions is reported. Valid expressions (which approximate the velocity maps of ODE systems) are evaluated at 50 points in $\mathbb{R}^d$ where each coordinate is uniformly sampled in $[-5, 5]$ (i.e. a Monte Carlo estimate) and the relative $L^2$ error is reported. Here $d$ is the dimension of the ODE system. More specifically, suppose $f(\mathbf{u})$ and $\hat{f}(\mathbf{u})$ are true and PROSE-generated ODE velocity maps, we report the average relative $L^2$ error computed at sampled points: $\frac{\|f - \hat{f}\|_2}{\|f\|_2}$.

## 4.1   RESULTS

We observe in Table 2 that all experiments, even those corrupted by noise or random terms, achieve low relative prediction errors ($< 5.7\%$). The data prediction error decreases as we relax the conditions on the symbolic guesses, i.e. when the equations are "Unknown" $5.7\%$ to "Known" $2.94\%$. Note in the case that the equations are "Known", we expect that the equations behave more like labels for the dataset. Moreover, the low expression error ($< 2.1\%$) shows PROSE's ability to correct and predict accurate equations, even when erroneous ODE equations are provided.

Table 2: **Performance of the model trained with different input expression types.** The two relative prediction errors are for interval $[2, 4]$ and $[2, 6]$, respectively.

| Experiments (Expression Type) | Relative Prediction Errors (%) | Relative Expression Error (%) | Percentage of Valid Expressions (%) |
|---|---|---|---|
| Known | 2.74, 2.94 | 0.00 | 100.00 |
| Skeleton | 3.39, 4.59 | 2.10 | 99.98 |
| Unknown (3D) | 3.43, 4.63 | 2.11 | 99.95 |
| Unknown (Multi-D) | 3.95, 5.66 | 1.88 | 99.94 |

**Data vs. Equation Prediction.**   We present the results of 10K testing samples in the "Unknown (3D)" experiment in Table 3. We see that the data prediction (whose features are influenced by the symbolic terms) is more accurate than using the learned governing equation directly. This shows the value of constructing a data prediction component rather than only relying on the learned governing equation. However, as in Kamienny et al. (2022), the predicted equations can be further refined

using optimization techniques, typically Broyden–Fletcher–Goldfarb–Shanno (BFGS) algorithm, where the predicted expression parameters can be used as a close initial guess.

Table 3: **Performance of data decoder output and symbol decoder output plus the backward differentiation formula (BDF method).**

| Prediction Generation Method | Relative Prediction Error (%) | Percentage of Valid Expression Outputs (%) |
|---|---|---|
| Data decoder output | 4.59 | 99.96 |
| Symbol decoder output + BDF method | 14.69 | |

**Out-of-distribution Case Study.** We study our model's ability to generalize beyond the training distribution. Specifically, we test on datasets whose parameters are sampled from a large interval $[F - \lambda F, F + \lambda F]$, where $F$ represents a value of interest. We choose $\lambda = 0.15, 0.20$, which are greater than the training setting $\lambda = 0.10$. The results are shown in Table 4. This shows that the approach can be used for prediction even in the case where the parameter values were not observed during training time.

Table 4: **Out-of-distribution Testing Performance.** The trained model is from the Unknown (3D) experiment. Relative prediction errors are reported for intervals $[2, 4]$ and $[2, 6]$, respectively.

| Parameter Sample Relative Range $\lambda$ | Relative Prediction Errors (%) | Relative Expression Error (%) | Percentage of Valid Expression Outputs (%) |
|---|---|---|---|
| 0.10 | 3.43, 4.63 | 2.11 | 99.95 |
| 0.15 | 3.89, 5.71 | 3.21 | 99.44 |
| 0.20 | 4.94, 7.66 | 4.83 | 97.89 |

**Ablation Study.** Since the model is multimodal in both inputs and outputs, we investigate the performance gains by using the equation embedding in the construction of the features. In particular, we compare the performance of the full PROSE model with multimodal input/output (as shown in Figure 2) and the PROSE model with only the data modality (i.e. no symbol encoder/decoder or fusion structures).

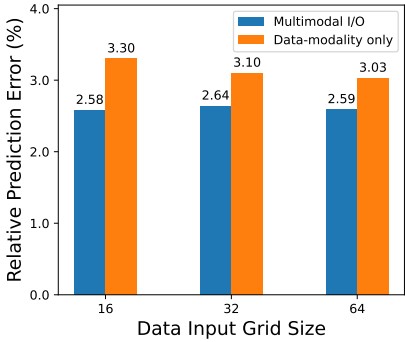
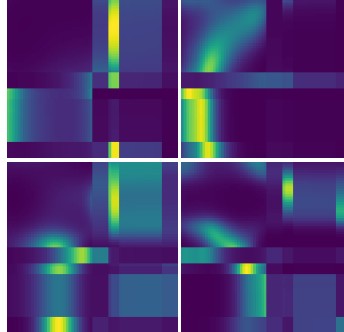

Figure 3: **Comparing the PROSE model with multimodal input/output and the PROSE model with only the data modality.** The models are trained with different data input lengths for 60 epochs. The relative prediction errors are computed on the same output grid.

Figure 4: **Sampled attention maps of feature fusion layers.** For each map, non-zero values in the upper left and bottom right corner represent in-modality interactions and non-zero values in the upper right and bottom left blocks represent cross-modality interactions. Other maps are presented in Appendix D.

The comparison tests are run using varying numbers of input sensors. For consistency, noise on the data samples is not included in this test, although the symbolic inputs do have unknown coefficients

and terms added/removed. As shown in Figure 3, the PROSE model with multimodal input/output consistently outperforms the data-modality-only model, demonstrating performance gains through equation embedding. Notably, we do not observe any degradation in the full PROSE model's performance when reducing the number of input sensors, whereas the data-modality-only model's performance declines as sensors are removed from the input function. This showcases the value of the symbol modality in supplying additional information for enhancing data prediction.

In Figure 4, we plot 4 (out of the $64 = 8$ layers $\times$ 8 heads) attention maps corresponding to the Feature Fusion layers on one four-wing attractor example (see Appendix B). This uses the full PROSE model with multimodal input/output and with a data input grid size 32. The non-zero values (which appear as the yellow/green pixels) indicate the connections between the features. More importantly, the non-zero values in the bottom-left and upper-right blocks indicate a non-trivial cross-modality interaction. Together with the improved relative error shown in Figure 3, we see the overall improvements using our multimodal framework.

**Output Example.** In Figure 5, we display a typical PROSE output from the "Unknown (3D)" experiment in Table 2. Each curve is one trajectory of one state variable $u_i(t)$ for $i = 1, 2, 3$. The target solution curves (with noise) are the dashed lines (only up to $t = 2$ is seen during testing) and the predicted solution curves are the solid lines. We display the target equation and the generated equation, which is exact with respect to the terms generated and accurate up to two digits (noting that the mantissa has length three).

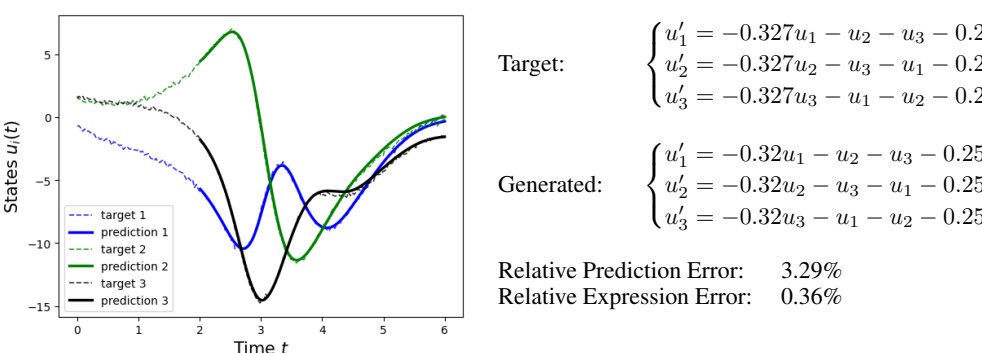

Target:
$$\begin{cases} u_1' = -0.327u_1 - u_2 - u_3 - 0.25u_2^2 \\ u_2' = -0.327u_2 - u_3 - u_1 - 0.25u_3^2 \\ u_3' = -0.327u_3 - u_1 - u_2 - 0.25u_1^2 \end{cases}$$

Generated:
$$\begin{cases} u_1' = -0.32u_1 - u_2 - u_3 - 0.25u_2^2 \\ u_2' = -0.32u_2 - u_3 - u_1 - 0.25u_3^2 \\ u_3' = -0.32u_3 - u_1 - u_2 - 0.25u_1^2 \end{cases}$$

Relative Prediction Error:     3.29%
Relative Expression Error:     0.36%

Figure 5: **An example of PROSE's outputs.** Target solution curves are dashed lines and predicted solution curves are solid lines. The input is the data up to $t = 2$. The numbers in the legend refer to the coordinate of the state variable $u_i(t)$ for $i = 1, 2, 3$. The target and PROSE generated equations are displayed.

## 5   DISCUSSION

The PROSE network is developed for model and multi-operator discovery. The network architecture utilizes hierarchical transformers to incorporate the data and embedded symbols in a symbiotic way. We show that the learned symbolic expression helps reduce the prediction error and provides further insights into the dataset. Experiments show that the generated symbolic expressions are mathematical equations with validity of $> 99.9\%$ on in-distribution tests and $> 97.89\%$ on out-of-distribution tests, and with numerical error of about $2\%$ (in terms of relative $L^2$ norm). This shows that the network is able to generate ODE models that correctly represent the dataset and does so by incorporating information from other similar ODEs.

The symbolic expression and data fusion yield a scientifically relevant multimodal formulation. In particular, the expressions provide alternative representation for the dataset and its predicted values, enabling the extraction of more refined information such as conserved quantities, stationary points, bifurcation regimes, hidden symmetries, and more. Additionally, since the symbolic expressions are valid functions, they can be used for evaluation and thus lead to alternative predictive algorithms (i.e. simulating the ODE). One future direction is the construction of a PROSE approach for nonlinear partial differential equations with spatio-temporal queries.

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

## A    EXPERIMENT SETUP

**Training**    A standard cross-entropy loss $\mathcal{L}_s$ is used for the symbolic outputs. While it is possible to simplify and standardize equations with `SymPy` (Meurer et al., 2017), d'Ascoli et al. (2022) showed that for their symbolic regression task, such simplification decreases training loss but not testing loss, thus we did not include it in our experiments.

Relative squared error $\mathcal{L}_d$ is used for the data predictions. In comparison to the standard mean squared error, the relative squared error makes the learning process more uniform across different types of ODE systems, as solution curves of different systems may have very different value ranges.

The data loss $\mathcal{L}_d$ and symbol loss $\mathcal{L}_s$ are combined to form the final loss function $\mathcal{L} = \alpha \mathcal{L}_d + \beta \mathcal{L}_s$, where the weights $\alpha, \beta$ are hyperparameters. Unless otherwise specified, the models are trained using the AdamW optimizer for 80 epochs where each epoch is 2,000 steps. On 2 NVIDIA GeForce RTX 4090 GPUs with 24 GB memory each, the training takes about 19 hours.

**Hyperparameters**    The model hyperparameters are summarized in Table 5, and the optimizer hyperparameters are summarized in Table 6.

Table 5: **Model hyperparameters.** FFN means feedforward network.

| Hidden dimension for attention | 512 | Hidden dimension for FFNs | 2048 |
|---|---|---|---|
| Number of attention heads | 8 | Fusion attention layers | 8 |
| Data encoder attention layers | 2 | Data decoder attention layers | 8 |
| Symbol encoder attention layers | 4 | Symbol decoder attention layers | 8 |

Table 6: **Optimizer hyperparameters.**

| Learning rate | $10^{-4}$ | Weight decay | $10^{-4}$ |
|---|---|---|---|
| Scheduler | Inverse square root | Warmup steps | 10% of total steps |
| Batch size per GPU | 256 | Gradient norm clip | 1.0 |
| Data loss weight $\alpha$ | 6.0 | Symbol loss weight $\beta$ | 1.0 |

## B    CHAOTIC AND MULTISCALE ODE DATASET

In this section, we provide the details of all ODE systems. We also include the parameters of interest.

**Thomas' cyclically symmetric attractor**

$$\begin{cases} u_1' & = \sin(u_2) - bu_1 \\ u_2' & = \sin(u_3) - bu_2 \\ u_3' & = \sin(u_1) - bu_3 \end{cases} \qquad b = 0.17$$

**Lorenz 3D system**

$$\begin{cases} u_1' & = \sigma(u_2 - u_1) \\ u_2' & = u_1(\rho - u_3) - u_2 \\ u_3' & = u_1 u_2 - \beta u_3 \end{cases} \qquad \begin{cases} \sigma = 10 \\ \beta = 8/3 \\ \rho = 28 \end{cases}$$

**Aizawa attractor**

$$\begin{cases} u_1' = (u_3 - b)u_1 - du_2 \\ u_2' = du_1 + (u_3 - b)u_2 \\ u_3' = c + au_3 - u_3^3/3 - u_1^2 + fu_3 u_1^3 \end{cases} \qquad \begin{cases} a = 0.95 \\ b = 0.7 \\ c = 0.6 \\ d = 3.5 \\ e = 0.25 \\ f = 0.1 \end{cases}$$

**Chen-Lee attractor**

$$\begin{cases} u_1' = au_1 - u_2 u_3 \\ u_2' = -10u_2 + u_1 u_3 \\ u_3' = du_3 + u_1 u_2/3 \end{cases} \qquad \begin{cases} a = 5 \\ d = -0.38 \end{cases}$$

**Dadras attractor**

$$\begin{cases} u_1' = u_2/2 - au_1 + bu_2 u_3 \\ u_2' = cu_2 - u_1 u_3/2 + u_3/2 \\ u_3' = du_1 u_2 - eu_3 \end{cases} \qquad \begin{cases} a = 1.25 \\ b = 1.15 \\ c = 0.75 \\ d = 0.8 \\ e = 4 \end{cases}$$

**Rössler attractor**

$$\begin{cases} u_1' = -u_2 - u_3 \\ u_2' = u_1 + au_2 \\ u_3' = b + u_3(u_1 - c) \end{cases} \qquad \begin{cases} a = 0.1 \\ b = 0.1 \\ c = 14 \end{cases}$$

**Halvorsen attractor**

$$\begin{cases} u_1' = au_1 - u_2 - u_3 - u_2^2/4 \\ u_2' = au_2 - u_3 - u_1 - u_3^2/4 \\ u_3' = au_3 - u_1 - u_2 - u_1^2/4 \end{cases} \qquad a = -0.35$$

**Rabinovich–Fabrikant equation**

$$\begin{cases} u_1' = u_2(u_3 - 1 + u_1^2) + \gamma u_1 \\ u_2' = u_1(3u_3 + 1 - u_1^2) + \gamma u_2 \\ u_3' = -2u_3(\alpha + u_1 u_2) \end{cases} \qquad \begin{cases} \alpha = 0.98 \\ \gamma = 0.1 \end{cases}$$

**Sprott B attractor**

$$\begin{cases} u_1' = au_2 u_3 \\ u_2' = u_1 - bu_2 \\ u_3' = c - u_1 u_2 \end{cases} \qquad \begin{cases} a = 0.4 \\ b = 1.2 \\ c = 1 \end{cases}$$

**Sprott-Linz F attractor**

$$\begin{cases} u_1' = u_2 + u_3 \\ u_2' = -u_1 + au_2 \\ u_3' = u_1^2 - u_3 \end{cases} \qquad a = 0.5$$

**Four-wing chaotic attractor**

$$\begin{cases} u_1' = au_1 + u_2u_3 \\ u_2' = bu_1 + cu_2 - u_1u_3 \\ u_3' = -u_3 - u_1u_2 \end{cases} \qquad \begin{cases} a = 0.2 \\ b = 0.01 \\ c = -0.4 \end{cases}$$

**Duffing equation**

$$\begin{cases} u_1' = 1 \\ u_2' = u_3 \\ u_3' = -\delta u_3 - \alpha u_2 - \beta u_2^3 + \gamma \cos(\omega u_1) \end{cases} \qquad \begin{cases} \alpha = 1 \\ \beta = 5 \\ \gamma = 8 \\ \delta = 0.02 \\ \omega = 0.5 \end{cases}$$

**Lorenz 96 system**

$$\begin{cases} u_i' = (u_{i+1} - u_{i-2})u_{i-1} - u_i + F, \ i = 1, \ldots, N \\ u_{-1} = u_{N-1}, \ u_0 = u_N, \ u_{N+1} = u_0 \end{cases} \qquad F = 8$$

**Double Pendulum**

$$\begin{cases} u_1' = u_3 \\ u_2' = u_4 \\ u_3' = \frac{-3g/l \sin(u_1) - g/l \sin(u_1 - 2u_2) - 2\sin(u_1 - u_2)(u_4^2 + u_3^2 \cos(u_1 - u_2))}{3 - \cos(2(u_1 - u_2))} \\ u_4' = \frac{\sin(u_1 - u_2)(4u_3^2 + 4g/l \cos(u_1) + u_4^2 \cos(u_1 - u_2))}{3 - \cos(2(u_1 - u_2))} \end{cases} \qquad \begin{cases} g = 9.81 \\ l = 1 \end{cases}$$

The initial conditions for the ODE systems are sampled uniformly from the hypercube $[-2, 2]^d$ where $d$ is the dimension of the system. The ODE systems are solved on the interval $[0, 6]$ using BDF method with absolute tolerance $10^{-6}$ and relative tolerance $10^{-5}$. Unless otherwise specified, the data part contains function values at 192 uniform grid points in the time interval $[0, 6]$, where the first 64 points in the interval $[0, 2]$ are used as data input points, and the last 128 points in the interval $[2, 6]$ are used as data labels. 2% Gaussian observation noise is added to the data samples. More precisely, if $u$ is the underlying true equation values, the observed value is $\tilde{u} = u + \sigma \eta$ where $\eta \sim \mathcal{N}(0, I)$ and $\sigma$ is chosen such that the signal-to-noise ratio $\frac{\sigma ||\eta||_2}{||u||_2}$ is 2%.

## C  ADDITIONAL RESULTS

In Table 7, we show the relative $L^2$ prediction error per equation type for all 3D equations. We also computed the dataset deviation (Lin et al., 2023) per equation type, which is the relative prediction error when the output is the average of all training examples of one equation class. The higher the dataset deviation, the more complicated the dataset (a model that simply predicts the average will have poor performance).

## D  VISUALIZATONS

Figure 6 shows the training and validation loss curves for experiments "Unknown (3D)" and "Skeleton" (described in Table 2). Figure 7 contains the attention architecture details. Figure 8 shows the full attention maps for one four-wing attractor example.

Table 7: **Per Equation-type Performance of the model in the experiment Unknown (3D).** The relative prediction error is for $[2, 6]$.

| Equation Type | # of Free Parameters | Relative Prediction Error (%) | Dataset Deviation (%) |
|---|---|---|---|
| Thomas' cyclically symmetric attractor | 1 | 2.51 | 99.99 |
| Lorenz 3D system | 3 | 2.64 | 41.98 |
| Aizawa attractor | 6 | 6.48 | 80.48 |
| Chen-Lee attractor | 2 | 3.56 | 100.00 |
| Dadras attractor | 5 | 9.79 | 105.29 |
| Rössler attractor | 3 | 2.61 | 99.99 |
| Halvorsen attractor | 1 | 5.80 | 83.21 |
| Rabinovich–Fabrikant equation | 2 | 4.87 | 100.04 |
| Sprott B attractor | 3 | 4.02 | 102.87 |
| Sprott-Linz F attractor | 1 | 4.14 | 193.87 |
| Four-wing chaotic attractor | 3 | 4.21 | 100.00 |
| Duffing equation | 5 | 4.87 | 58.88 |

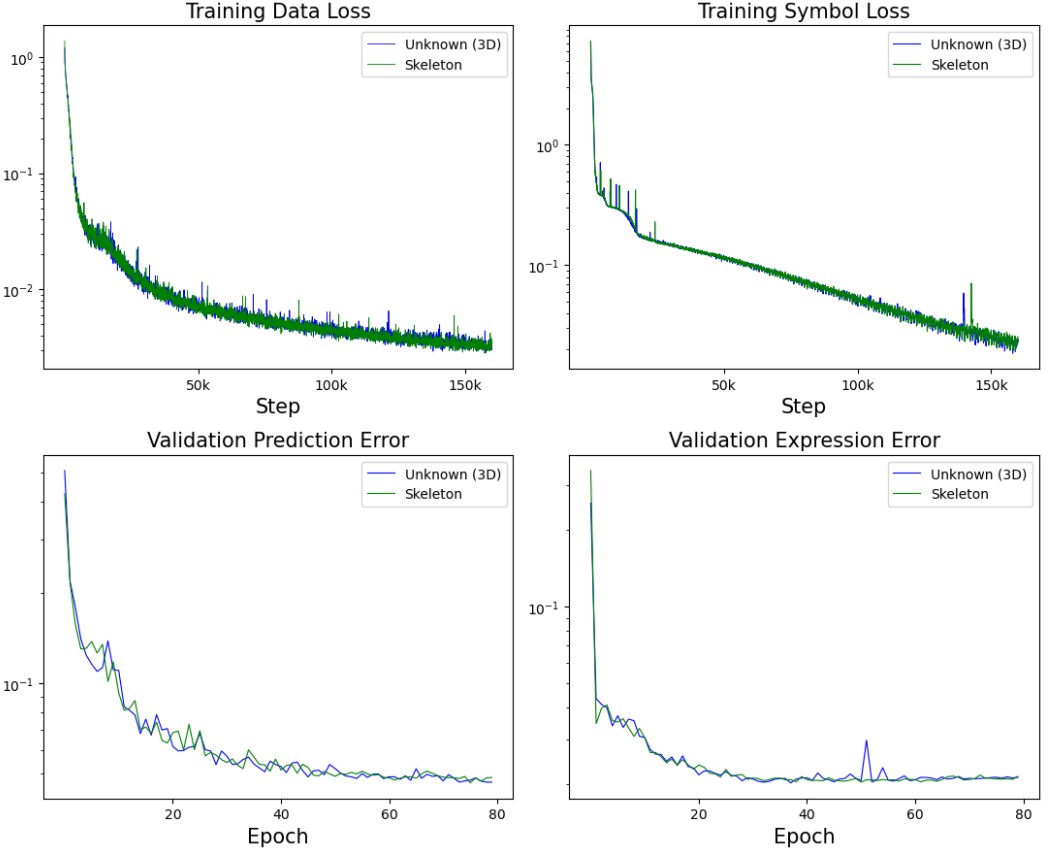

Figure 6: **Example training and validation loss curves.**

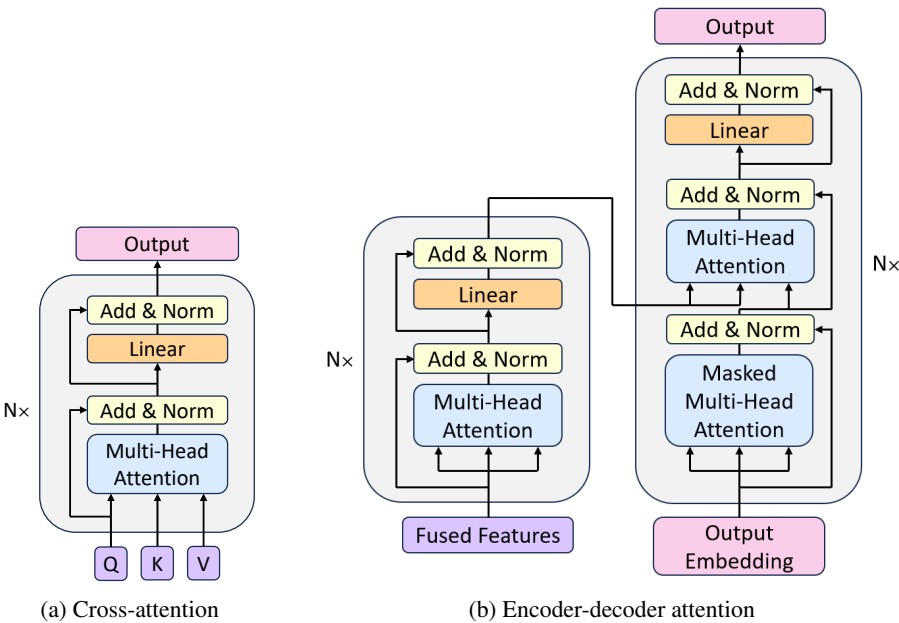

(a) Cross-attention      (b) Encoder-decoder attention

Figure 7: **Attention block details.** Self-attention is a special case of cross-attention with the same source.

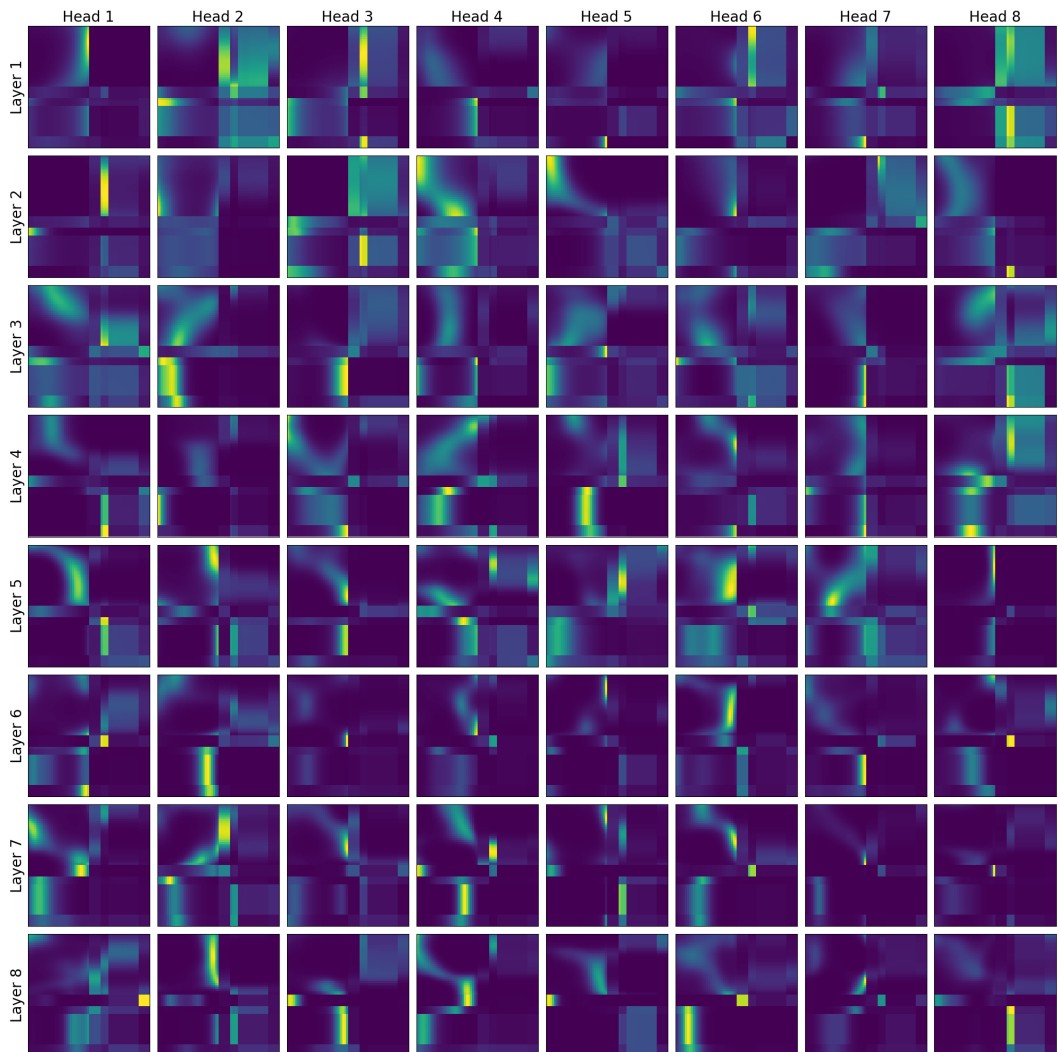

Figure 8: **Attention maps of 8 Feature Fusion layers for a four-wing attractor example.**

