# OpenReview forum: "PROSE: Predicting Operators and Symbolic Expressions using Multimodal Transformers"
_ICLR.cc/2024/Conference — Submitted to ICLR 2024_

### Official Review · Reviewer_45hW · 2023-10-12

**Soundness:** 3 good
**Presentation:** 3 good
**Contribution:** 3 good
**Rating:** 6
**Confidence:** 4

**Summary:**

This paper proposes Predicting Operators and Symbolic Expressions (PROSE) to learn from multimodal inputs to generate numerical predictions and mathematical equations. Experiments using 15 differential equations show highly precise results.

**Strengths:**

1. The research on differential equations using machine learning methods is of significant research interests.
2. A multi-modal approach using transformer architecture is technically sound, and the novelty and technical contribution is reasonable.
3. Experiments results are impressive with high precision.

**Weaknesses:**

1. There is no comparison with existing methods, so it is hard to assess how the proposed approach comparing with SOTA.
2. The paper claims that high performance partially comes from multimodal learning, an ablation study will be interesting to verify this claim.

**Questions:**

1. Can you compare with existing methods to verify that the high performance is not because the differential equations used are too simple?
2. An ablation study with single inputs will be interesting to show the effectiveness of multimodal input.

---

> ### Author Response · Authors · 2023-11-16
> **Responses to Reviewer 45hW**
>
> Dear Reviewer 45hW,
>
> Thank you for the valuable feedback. Below we address your questions.
>
> ---
>
> >  **Q1.** Can you compare with existing methods to verify that the high performance is not because the differential equations used are too simple?
>
> **A1.** In this paper, we are proposing a new model for the new task of training a single neural network to handle forward/inverse problems for multiple types of equations, or, we construct a neural network to handle multi-operator learning. As a comparison, popular operator learning frameworks (e.g. FNO, DeepONet) can only handle one type of equation or operator.
>
> To the best of our knowledge, PROSE is the first and only method that tackles the multi-operator learning problem with vague equation labels. However, as you correctly asked, an ablation study comparing different variants of the PROSE model is necessary. An ablation study was included in the original submission.
>
> > **Q2.** An ablation study with single inputs will be interesting to show the effectiveness of multimodal input.
>
> **A2.** The ablation study on single vs multimodal inputs appears in section 4.1 (page 8), where we compared the performance of the full PROSE model with multimodal input (system solution with equation encoding) and the model with only the data modality (system solution). This was in the original submission.
>
> We can observe from Figure 3 that, as we reduce the data modality information to the network, the full PROSE model is more robust than data only. However, the accuracy of the data-only variant decreases as we reduce the size of the data input.

---

> > ### Comment · Reviewer_45hW · 2023-11-22
> > **Thanks for the authors' response.**
> >
> > Based on the authors' response, my rating remains the same.

---

### Official Review · Reviewer_LoCo · 2023-10-29

**Soundness:** 2 fair
**Presentation:** 3 good
**Contribution:** 2 fair
**Rating:** 5
**Confidence:** 4

**Summary:**

The paper proposes PROSE (Predicting Operators and Symbolic Expressions), a novel approach to predict the evolution of a solution of a certain multi-dimensional ODE, given the first time-steps of the solution and symbolic information about the velocity field of the ODE.

Alongside the numerical prediction of the ODE's solution at future time steps, the model also outputs a symbolic expression representing the velocity field of the underlying ODE.

The method is based on fusing multi-modal input information in the form of numerical data (ODE's solution sampled up to a certain time step with noise perturbations) and a symbolic, potentially wrong or corrupted, representation of the velocity field. PROSE is entirely based on Transformers and the attention mechanism and draws inspiration from the pioneering line of work on operator learning for differential equations.

The experiments, performed on 15 multi-dimensional ODE systems, show that the model efficiently leverages information from both the numerical and symbolic domains and predicts accurate solutions both at the numerical and the symbolic levels.

**Strengths:**

- The paper is generally well-written and background information is carefully provided.

- The topic of merging and combining numerical and symbolic information in Transformer models is very interesting and the paper takes a  relevant step in this direction.

- The experiments effectively show that the model is able to leverage symbolic information in the task of predicting the future time steps of the ODE's solution. This means that symbolic information is successfully combined with numerical information thanks to the modality fusion step.

**Weaknesses:**

- My main concern is that the claims about the ability of PROSE to perform model discovery are not sufficiently backed by empirical evidence. The number of considered ODEs is limited to 15 and as such, I do not see the necessity of having a symbolic decoder predicting the full mathematical expression. Alternatively, one could have resorted to a simple classification network responsible for predicting one out of 15 classes corresponding to the underlying ODE. The coefficient of the predicted ODE could have then been optimized via, for example, BFGS. In other words, I feel that the hypothesis space considered in the paper is too small, as the number of possible expressions to be predicted is limited to 15.

- Another source of concern is that the model is trained on fixed grids, i.e. training solutions are provided up to 2 seconds. It would be interesting to see how the performance of the model changes as the size of the training window changes.

- I think a more complete analysis should have been performed for the case when input symbolic information is wrong or corrupted by noise. Is the symbolic decoder able to correct the wrong information provided by the symbolic encoder?

- In terms of novelty, while I understand that fusing symbolic and numerical information can help in the prediction of the solution at future time steps, when symbolic information is not available (which is rather common), the model reduces to a relatively simple forecasting approach, whose novelty is quite limited. What would make the approach more novel is the symbolic decoder part, which in the considered setting of only 15 equations, cannot do much more than predicting one of the 15 examples seen at training time.

- Some relevant related works are not mentioned. In particular [1,2] also use Transformer to predict velocity fields at the symbolic level.

- The model is not compared with relevant baselines, e.g. SINDy or DeepOnet, or the aforementioned works.


[1] Becker, S., Klein, M., Neitz, A., Parascandolo, G., & Kilbertus, N. (2023). Predicting Ordinary Differential Equations with Transformers.

[2] d'Ascoli, Stéphane, et al. "ODEFormer: Symbolic Regression of Dynamical Systems with Transformers." arXiv preprint arXiv:2310.05573 (2023).

**Questions:**

See weaknesses part.

---

> ### Author Response · Authors · 2023-11-16
> **Responses to Reviewer LoCo (1/2)**
>
> Dear Reviewer LoCo,
>
> Thank you for the valuable feedback. Below we address your questions.
>
> ---
>
> > **Q1.** My main concern is that the claims about the ability of PROSE to perform model discovery are not sufficiently backed by empirical evidence. The number of considered ODEs is limited to 15 and as such, I do not see the necessity of having a symbolic decoder predicting the full mathematical expression. Alternatively, one could have resorted to a simple classification network responsible for predicting one out of 15 classes corresponding to the underlying ODE. The coefficient of the predicted ODE could have then been optimized via, for example, BFGS. In other words, I feel that the hypothesis space considered in the paper is too small, as the number of possible expressions to be predicted is limited to 15.
>
> **A1.** Thanks for your questions. We would like to emphasize that our task focuses on training a single network capable of constructing the solution operators for multiple different dynamical systems simultaneously with generating or completing the full equation model. One of the purposes of using symbolic expressions is to encode the equations which enable the model to identify different operators. The key object is to construct the family of operators for multiple systems.
>
> During the training process, we intentionally remove some terms or add erroneous terms to the true system, which significantly enlarges the space of the total ODEs.
> Moreover, as we focus on operator construction, we calculate the dataset deviation (i.e. we use the mean over all training sampling solutions as the prediction for all testing solutions and calculate the relative error), one potential metric in the operator learning society to determine the difference between the testing and training dataset (reference in main text). The deviations range from about 43% to 194% which shows that the testing and training samples substantially deviate from each other.
>
> The symbolic decoder is not doing classification or model discovery from scratch, but doing inverse problems and corrections to the input equation guesses. It provides a flexible way of solving inverse problems with different numbers of free parameters. For discrete Markovian sequences, using BFGS to optimize the parameters gives good performance improvement ([3]). However, for dynamical systems, using BFGS (and other Newton solvers) are sensitive to initial guesses and can be unstable due to time stepping.
>
> In regards to the dataset, for 15 classes of ODEs, we are randomly sampling the free parameters, resulting in a much larger collection of forward/inverse operators than just 15. As a comparison, in [2] (which focuses on the pure symbolic regression task), their testing dataset contains 63 equations where only the initial conditions are sampled. For randomly generated velocity fields, there is a high probability that the solution curves either provide redundant information (e.g. quickly converging to an invariant set) or are not well-defined (e.g. finite time singularities.)
>
> > **Q2.** Another source of concern is that the model is trained on fixed grids, i.e. training solutions are provided up to 2 seconds. It would be interesting to see how the performance of the model changes as the size of the training window changes.
>
> **A2.** Single operator learning learns the mapping between two function spaces. The variable training window size is a challenging problem whose input functions may not share the domain.
> To the best of our knowledge, this requires the input functions to share the domain.
> This is only being addressed very recently, see [5] which uses the distributed learning idea to relax the "same input function domain" assumption. However, the method in [5] works for some subsets of the entire dataset.
> Our work is for multi-operator learning, which adds many additional layers of complexity over the single operator learning approaches.
> It is possible to formulate the problem with a variable window size using the reference work (extending to the multi-operator learning setting); however, we will leave this as a future direction.
>
> > **Q3.** I think a more complete analysis should have been performed for the case when input symbolic information is wrong or corrupted by noise. Is the symbolic decoder able to correct the wrong information provided by the symbolic encoder?
>
> **A3.** In our original submission, the "Unknown" experiments in Table 1 discuss this case. Specifically, all numerical coefficients are erased, terms are randomly deleted from the true systems, and erroneous terms are randomly added. So the input guesses are corrupted and noisy. The low relative expression errors in Table 2 show that the symbolic decoder can correct wrong information.

---

> ### Author Response · Authors · 2023-11-16
> **Responses to Reviewer LoCo (2/2)**
>
> > **Q4.** In terms of novelty, while I understand that fusing symbolic and numerical information can help in the prediction of the solution at future time steps when symbolic information is not available (which is rather common), the model reduces to a relatively simple forecasting approach, whose novelty is quite limited. What would make the approach more novel is the symbolic decoder part, which in the considered setting of only 15 equations, cannot do much more than predicting one of the 15 examples seen at training time.
>
> **A4.** PROSE is solving the multi-operator learning problem. Specifically, one needs to learn the mapping from the input function together with the operator information to the output function. This is a non-trivial problem. Even without the symbolic and fusion parts, PROSE is not a seq2seq model and not a "simple forecasting approach". The data decoder for PROSE is an operator network that can be evaluated at arbitrary time-points. This mitigates the accumulation of error that can be readily seen in seq2seq predictions of dynamical systems with growth or chaotic behavior ([4]). This allows PROSE to handle longer predicting windows without exponential error growth. In addition, using seq2seq Transformer models results in quadratic complexity (in the length of the prediction window), while our model has linear complexity (in the length of the prediction window).
>
> In regards to the symbolic aspect, the equation information enhances the accuracy of the multi-operator learning framework as this will enable the network to know which equation or system will be used. As a result, the approach benefits from including system information in the network. Our work uses symbolic expressions to encode the operator and equation, that information has to be used to inform the network of the operator learning to be constructed. This is detailed in the "Unknown" setting in Table 1. In particular, the PROSE model returns a consistent relative error (3.95% and 5.66%) even when information is severely degraded. Now, if the task were for a single operator, then it would be the case that the symbolic information is not necessary, but is not the problem addressed in the work.
>
> > **Q5.** Some relevant related works are not mentioned. In particular [1,2] also use Transformer to predict velocity fields at the symbolic level.
>
> **A5.** Thank you for pointing this out. Please note that paper [2] did not appear online until after the submission deadline for ICLR 2024, thus it would not have been possible for us to mention that at the time of this submission. We now include [1] in the related works section.
>
> > **Q6.** The model is not compared with relevant baselines, e.g. SINDy or DeepOnet, or the aforementioned works.
>
> **A6.** PROSE is a multi-modal input, multi-modal output model. Specifically, the PROSE model constructs the operator and learns equations for multiple distinct systems simultaneously. To the best of our knowledge, this is the first work addressing this task.
>
> SINDy and related works learn the equation but not the operator and require a fixed dictionary of possible equations. DeepONet learns a single operator (which is not suitable for multi-operator learning tasks) and does not generate an explicit model or equation. A direct comparison to either SINDy or DeepONet would not be fair to those approaches (these are different tasks).
>
> We also would like to emphasize that the motivation for multimodality stems from the need for equation information to enhance multi-operator learning. This is verified in the "Ablation Study section" of our work. Particularly, we gradually reduce the information required for operator construction, however, due to the presentation of the equation information and the guidance from the equation learning loss, the operator learning relative error does not decrease.
>
> ---
>
> **References**
>
> [1] Becker, S., Klein, M., Neitz, A., Parascandolo, G., & Kilbertus, N. (2023). Predicting Ordinary Differential Equations with Transformers.
>
> [2] d'Ascoli, Stéphane, et al. (2023). ODEFormer: Symbolic Regression of Dynamical Systems with Transformers. arXiv preprint arXiv:2310.05573.
>
> [3] Kamienny, P. A., d'Ascoli, S., Lample, G., & Charton, F. (2022). End-to-end symbolic regression with transformers. Advances in Neural Information Processing Systems, 35, 10269-10281.
>
> [4] Qin, T., Chen, Z., Jakeman, J. D., & Xiu, D. (2021). Data-driven learning of nonautonomous systems. SIAM Journal on Scientific Computing, 43(3), A1607-A1624.
>
> [5] Zhang, Z., Moya, C., Lu, L., Lin, G., & Schaeffer, H. (2023). D2NO: Efficient Handling of Heterogeneous Input Function Spaces with Distributed Deep Neural Operators. arXiv preprint arXiv:2310.18888.

---

### Official Review · Reviewer_62Ac · 2023-10-30

**Soundness:** 2 fair
**Presentation:** 2 fair
**Contribution:** 2 fair
**Rating:** 5
**Confidence:** 4

**Summary:**

The authors propose a bimodal transformer-based model to predict solutions of (partly) known ordinary differential systems from a few initial points in their trajectories.

The initial points on the trajectories to be predicted are encoded by a two-layer transformer. The system equations, which be fully known, known up to their pre-factors or known up to a few random terms, are represented as trees, enumerated in Polish notation, and encoded by a 4-layer transformer. The output of both encoders are then mixed together by a 8-layer transformer.

Two decoders (both 8-layer transformers) operate on the encoded inputs: one to predict future values of the trajectories from their time coordinates, and one (auto-regressive) to predict the system equation.

Experiments on 15 dynamical systems, with parameters sampled in a small range around a value of interest, show that the model can indeed predict trajectories and actual model parameters with good accuracy, and that the model can extrapolate away from the equation parameters seen at train time.

**Strengths:**

Using bimodal numeric/symbolic models is a promising approach, and the architecture proposed makes a lot of sense. The initial results are interesting and promising.

**Weaknesses:**

The methodology must be better justified. The overall architecture is fairly complex, and many technical choices and not justified, or validated by ablation studies. Some important information, like the training loss, is missing.

The training and test sets are too small and not diverse enough. All train and test examples are generated from only 15 systems, with a very small number of free parameters (4/15 have 1 free parameter, 3/15 have 2 and 4/15 have 3). Those parameters are  sampled in a small interval (0.9,1.1) around their value of interest, to generate a training sample of 512k examples -- i.e. 34,000 examples per system. There is a large risk that the training and test equations will significantly overlap, or at least be very similar. What guarantees do you have that your results are not due to train/test contamination?

The experiments are the weak part of the paper. Appendix B states that the model was run for 80 epochs of 2000 optimisation steps, on batches of 2*256=512 examples. Overall, this means 81.9 million training examples, or 160 passes on the 512k training set. Given the size of the model (several hundred million parameters), and the lack of diversity in the train set (34k examples per system on average), there is a large risk that the model memorize its training data. This might account for some of the results.

Finally, a comparison with baseline results, notably prior work on symbolic regression(see for instance, https://arxiv.org/abs/2307.12617), would help assess put the results in perspective. In their current form, the benefits of this approach are difficult to evaluate.

Overall, this paper is interesting and the model shows promise but stronger experimental evidence, and especially a larger and more diverse training set, are needed to validate its results.

**Questions:**

* related works: previous works on symbolic regression should be mentioned, notably Becker et al. (ICML 2023, https://arxiv.org/abs/2307.12617), and probably D'Ascoli et al., Biggio and Kamienny et al.
* Polish notation: in language models, it was introduced by Lample & Charton in 2019, but the technique is much older (logicians in the 1930s, and computer scientists in the 1950s)
* "there is a one-to-one correspondence between trees and mathematical expressions" this is incorrect: expressions like x-2.1 can be encoded as + x -2.1 or - x 2.1, and x+y+z as + +x y z or + x + y z. Besides, unless simplification is used when the data is prepared, many equivalent expressions result in different trees (2+x <=> x+2 <=> 1+x+1).
* the 3-token encoding for floating point numbers was introduced in Charton 2022. Lample et al. only use integer pre-factors.
* "our vocabulary is also of order $10^4$ words." What precision do you use? Charton 2022 uses 3 significant digits, for a vocabulary size of about 1100 words (901 mantissas, 200 exponents, 2 signs). D'Ascoli 2023 uses 4, for a vocabulary a little below $10^4$.
* Figure 2: the term self-attention layer used in the figure is misleading: these are self-attention+FFN layers, with most of the trainable parameters in the FFN. Maybe use transformer layer instead, or transformer encoder and transformer decoder to indicate the presence of the cross-attention mechanism.
* feature fusion: an 8-layer transformer seems like a very large model to learn to align the numeric and symbolic representations learned up-front. Have you tried smaller fusion networks (one or two layers, perhaps just self-attention and a single linear layer as the output?)
* since the symbolic output can have variable length, many authors propose to compress it as a single, high dimensional vector, using attention distillation (Santos 2016), or simpler techniques like max-pool. Have you tried such techniques?
* what is the training loss? how do you balance between the two decoders, are they trained simultaneously, or separately?
* could you provide results as a function of system dimension, and number of free parameters?
* could other metrics be presented? (D'Ascoli uses $L^\infty$, Charton $L^1$)
* Table 4, seems to be on the Unknown 3D experiments, please add this to the caption.
* in table 2, a study of how error increases with the extrapolation interval would be interesting: from data sampled on (0,2) what is the error on (2,4), (4,6), (6,8)?

---

> ### Author Response · Authors · 2023-11-16
> **Responses to Reviewer 62Ac (1/2)**
>
> Dear Reviewer 62Ac,
>
> Thank you for the valuable feedback. Below we address your questions.
>
> ---
>
> > **Q1.** Related works: previous works on symbolic regression should be mentioned, notably Becker et al. (ICML 2023, https://arxiv.org/abs/2307.12617), and probably D'Ascoli et al., Biggio and Kamienny et al.
>
> **A1.** We would like to emphasize that our task is to train a single network capable of constructing solution operators for a mix of forward/inverse problems for multiple types of equations. This is a simultaneous task. The two decoders are for the two downstream tasks: the data decoder for the forward problem, and the symbol decoder for the inverse problem. The task is very different from (pure) symbolic regression and is more related to a new multi-operator learning task that utilizes symbolic regression. The symbolic guess inputs for our task serve as "vague" labels for the model as well as templates for the symbol decoder to solve inverse problems. The representation of symbolic expressions is similar to the works mentioned by the reviewer, which we properly cited in section 3. We added additional reference to symbolic regression approaches related to learning equations from data, since that is more closely related to the multi-operator approach we proposed.
>
> > **Q2.** Polish notation: in language models, it was introduced by Lample & Charton in 2019, but the technique is much older (logicians in the 1930s, and computer scientists in the 1950s)
>
> **A2.** Thank you for addressing this. We added a citation to the English translation of the original Polish/Prefix notation work from 1931.
>
> > **Q3.** "There is a one-to-one correspondence between trees and mathematical expressions." This is incorrect: expressions like x-2.1 can be encoded as + x -2.1 or - x 2.1, and x+y+z as + +x y z or + x + y z. Besides, unless simplification is used when the data is prepared, many equivalent expressions result in different trees (2+x <=> x+2 <=> 1+x+1).
>
> **A3.** The cited sentence should not be taken out of context. The full sentence is "Under some additional restrictions (e.g. 1 + 2 + 3 should be processed as 1 + (2 + 3), -1 $\times$ x is equivalent to -x), there is a one-to-one correspondence
> between trees and mathematical expressions," which addresses the reviewer's concerns since this is conditioned on additional restrictions.
>
> > **Q4.** The 3-token encoding for floating point numbers was introduced in Charton 2022. Lample et al. only use integer pre-factors.
>
> **A4.** Thank you for pointing this out. The citation has been updated.
>
> > **Q5.** "Our vocabulary is also of order $10^4$ words." What precision do you use? Charton 2022 uses 3 significant digits, for a vocabulary size of about 1100 words (901 mantissas, 200 exponents, 2 signs). D'Ascoli 2023 uses 4, for a vocabulary a little below $10^4$.
>
> **A5.** Thank you for pointing this out. This is a typo and has been fixed in the updated paper. We use 3 significant digits and our vocabulary also contains about 1100 words.
>
> > **Q6.** Figure 2: the term self-attention layer used in the figure is misleading: these are self-attention+FFN layers, with most of the trainable parameters in the FFN. Maybe use transformer layer instead, or transformer encoder and transformer decoder to indicate the presence of the cross-attention mechanism.
>
> **A6.** Thank you for addressing the potential confusion. The main reason behind the naming convention is that there are three different attention mechanisms used in the model: self-attention, cross-attention, and encoder-decoder attention. We have updated the captions to reflect the feedforward networks.
>
> > **Q7.** Feature fusion: an 8-layer transformer seems like a very large model to learn to align the numeric and symbolic representations learned up-front. Have you tried smaller fusion networks (one or two layers, perhaps just self-attention and a single linear layer as the output?)
>
> **A7.** Thank you for the insightful question. For our multi-operator learning task, as there are two input modalities, the feature fusion part is the backbone where the construction of forward/inverse solution operators happens. Since a larger portion of the impactful learning occurs in the fusion layer, this is reflected in the number of attention layers. This is another key difference from (pure) symbolic regression, where decoders are often the backbone. The number of attention layers for each component is a hyperparameter which we choose based on a balance of model size and performance.

---

> ### Author Response · Authors · 2023-11-16
> **Responses to Reviewer 62Ac (2/2)**
>
> > **Q8.** Since the symbolic output can have variable length, many authors propose to compress it as a single, high-dimensional vector, using attention distillation (Santos 2016), or simpler techniques like max-pool. Have you tried such techniques?
>
> **A8.** Thanks for your suggestions. We use auto-regressive prediction to perform input equation correction and inverse problems. We use the dimension-separation token and end-of-sentence token to switch to another dimension and stop the prediction, respectively.  We were not able to locate the paper mentioned "Attention Distillation (Santos 2016)."
>
> > **Q9.** What is the training loss? How do you balance between the two decoders, are they trained simultaneously, or separately?
>
> **A9.** The training process and loss appear in Appendix A of the original submission. The two decoders are trained simultaneously where the final loss is a linear combination of losses for the two decoders. The coefficients are hyperparameters determined through coarse grid search.
>
> > **Q10.** Could you provide results as a function of system dimension, and number of free parameters?
>
> **A10.** We created Table 7 in Appendix C detailing the error as a function of the number of free parameters.
>
> > **Q11.** Could other metrics be presented? (D'Ascoli uses $L^\infty$, Charton $L^1$)
>
> **A11.** Our task focuses on a new operator learning approach, where the relative $L^2$ error is the standard metric.
>
> > **Q12.** Table 4, seems to be on the Unknown 3D experiments, please add this to the caption.
>
> **A12.** Thank you for pointing this out. We have updated the caption for Table 4 to reflect this.
>
> > **Q13.** In Table 2, a study of how error increases with the extrapolation interval would be interesting: from data sampled on (0,2) what is the error on (2,4), (4,6), (6,8)?
>
> **A13.** In table 2, we are providing relative $L^2$ error for the interval $[2,4]$ and $[2,6]$. The extrapolation regime is 2x the training interval. A full extrapolation study is left for future direction.
>
> ---
>
> **Response to the "weakness" part:**
>
> Multi-operator learning is the key question we address in this work. We use the symbols as the encoding of the operator and to enable the network to understand and identify different operators. As the goal is to construct the operator, we focus more on the extrapolation of constructing the solution operators. To show that the training and testing sets are dissimilar, we calculate the dataset deviation per equation type. That is, we compute the mean of all training sample solutions and use the mean as the prediction for all testing samples. As shown in Table 7, these values range from 42% to 194%, which shows the testing samples deviate significantly from the training sample.
>
> Each type of ODE contains a family of equations, i.e. the parameters are sampled from a distribution. To better test the extrapolation property, we study the out-of-distribution predictions. As detailed in the Out-of-distribution Case Study Section, we test parameters outside of the training range. The errors presented in Table 4 show that the PROSE model indeed learns the operator instead of memorization.
>
> In regards to train/test contamination: our task is different from symbolic regression in that the model doesn't need to generate equations from scratch.
>
> In additional regards to memorization: The model has 100 million parameters. As you computed, each type of equation has about 34k training examples.  Although there are 15 distinct families of ODEs, since we randomize the parameters (which could result in different phenomenological behaviors) there are 1.7k different sets of training equations (this is a direct way to compare to single operator learning). Given that the solutions of chaotic systems are sensitive with respect to their parameters and initial data, the training dataset contains a rich collection of solution curves with different behaviors. The out-of-distribution testing (in section 4) also shows that the model is not just memorizing the training dataset.

---

> > ### Comment · Reviewer_62Ac · 2023-11-22
> > **Thank you for your response**
> >
> > Thank you very much for you replies, which clarify several of my concerns. I still stand by my comment that the paper needs additional experimental evidence for the method to prove its worth.
> >
> > I am happy to raise my rating from 3 to 5.
> >
> > Note: here is the reference to the Santos paper: https://arxiv.org/abs/1602.03609

---

### Author Response · Authors · 2023-11-16
**General Responses**

Dear AC and Reviewers,

Thank you for your valuable feedback. Below are some general responses given the comments and concerns from the reviewers.

Here is a summary of edits we made to the revised manuscript:

- An added Appendix C and Table 7 displaying the dataset deviations (difference between training and testing dataset) and prediction errors for each type of equation.

- Added citations to the relevant symbolic regression works in Related Works (Section 2), based on reviewers' feedback.

- Minor edits (e.g. caption updates, typo corrections).

We would like to emphasize that the main task is to train a single network capable of constructing solution operators for a mix of forward/inverse problems for multiple types of equations and to generate the full model using partial guesses. The proposed work is the first to address the multi-operator learning and multi-modal output task. The task is very different from (pure) symbolic regression and is more related to a new multi-operator learning task that utilizes symbolic regression. This is also distinct from single operator learning, which trains a solution operator for one type of equation.

To show that the training and testing sets are dissimilar, we calculate the dataset deviation for each type of equation where the mean of the training samples is used. As shown in the newly added Table 7, these values range from 42% to 194%, which shows the testing samples deviate significantly from the training sample.

Each type of ODE contains a family of equations, i.e. the parameters are randomly sampled. In the Out-of-distribution Case Study section, we test parameters outside of the training range. The errors presented in Table 4 show that the PROSE model indeed learns the operator instead of memorization.

Additionally, although there are 15 distinct families of ODEs, since we randomize the parameters (which could result in different phenomenological behaviors) there are 1.7k different sets of training equations (this is a direct way to compare to single operator learning). Given that the solutions of chaotic systems are sensitive with respect to their parameters and initial data, the training dataset contains a rich collection of solution curves with different behaviors. The out-of-distribution testing (in section 4) also shows that the model is not just memorizing the training dataset.

---

### Meta-Review · Area_Chair_5ARN · 2023-12-10

**Metareview:**

This paper proposes PROSE (Predicting Operators and Symbolic Expressions) a transformer-based method that maps bimodal inputs to bimodal outputs in order to predict the evolution of a solution of an ordinary differential equation (ODE) given the first few time-steps of the solution and symbolic information about the velocity field of the ODE. The output of the model is (1) the numerical prediction of the ODE’s solution at future time steps and a symbolic expression representing the velocity field of the underlying ODE.

All the reviewers agree that the use of bimodal information in the proposed method is a promising approach and that the proposed architecture makes a lot of sense. I agree with the reviewers, and I commend the authors for this. However, the empirical evaluation was a common concern across the reviewers and it is hard to ignore this. In particular, the reviewers believe that the claimed contributions are not quite backed up by empirical evidence as there were major concerns on generalization given the limited set of differential equations (only 15 systems). The reviewers also believed that many technical choices were not justified. Finally, comparison with other approaches must also be investigated. The authors response mentioned that previous work has not considered their setting but I think one can still compare to previous approaches suggested by the reviewers under constrained settings, for example, when predicting a single output (unimodal), regardless of what the system uses.

**Justification For Why Not Higher Score:**

It is a good promising idea but not sufficiently demonstrated by experiments.

**Justification For Why Not Lower Score:**

N/A

---

### Decision · Program_Chairs · 2024-01-16

Reject